# Evaluation and Validation of the Limited Sampling Strategy of Polymyxin B in Patients with Multidrug-Resistant Gram-Negative Infection

**DOI:** 10.3390/pharmaceutics14112323

**Published:** 2022-10-28

**Authors:** Xueyong Li, Bingqing Zhang, Yu Cheng, Maohua Chen, Hailing Lin, Binglin Huang, Wancai Que, Maobai Liu, Lili Zhou, Qinyong Weng, Hui Zhang, Hongqiang Qiu

**Affiliations:** 1College of Pharmacy, Fujian Medical University, Fuzhou 350004, China; 2Department of Pharmacy, Fujian Medical University Union Hospital, Fuzhou 350001, China; 3Department of Critical Care Medicine, Fujian Medical University Union Hospital, Fuzhou 350001, China

**Keywords:** polymyxin B, limited sampling strategy, multiple linear regression, validation, therapeutic drug monitoring

## Abstract

Polymyxin B (PMB) is the final option for treating multidrug-resistant Gram-negative bacterial infections. The acceptable pharmacokinetic/pharmacodynamic target is an area under the concentration–time curve across 24 h at a steady state (AUC_ss,24h_) of 50–100 mg·h/L. The limited sampling strategy (LSS) is useful for predicting AUC values. However, establishing an LSS is a time-consuming process requiring a relatively dense sampling of patients. Further, given the variability among different centers, the predictability of LSSs is frequently questioned when it is extrapolated to other clinical centers. Currently, limited data are available on a reliable PMB LSS for estimating AUC_ss,24h_. This study assessed and validated the practicability of LSSs established in the literature based on data from our center to provide reliable and ready-made PMB LSSs for laboratories performing therapeutic drug monitoring (TDM) of PMB. The influence of infusion and sampling time errors on predictability was also explored to obtain the optimal time points for routine PMB TDM. Using multiple regression analysis, PMB LSSs were generated from a model group of 20 patients. A validation group (10 patients) was used to validate the established LSSs. PMB LSSs from two published studies were validated using a dataset of 30 patients from our center. A population pharmacokinetic model was established to simulate the individual plasma concentration profiles for each infusion and sampling time error regimen. Pharmacokinetic data obtained from the 30 patients were fitted to a two-compartment model. Infusion and sampling time errors observed in real-world clinical practice could considerably affect the predictability of PMB LSSs. Moreover, we identified specific LSSs to be superior in predicting PMB AUC_ss,24h_ based on different infusion times. We also discovered that sampling time error should be controlled within −10 to 15 min to obtain better predictability. The present study provides validated PMB LSSs that can more accurately predict PMB AUC_ss,24h_ in routine clinical practice, facilitating PMB TDM in other laboratories and pharmacokinetics/pharmacodynamics-based clinical studies in the future.

## 1. Introduction

Multidrug-resistant (MDR) Gram-negative bacteria, including Enterobacteriaceae, *Acinetobacter baumannii*, and *Pseudomonas aeruginosa*, have rapidly developed worldwide. In particular, carbapenem-resistant Enterobacteriaceae are a type of nosocomial pathogen that is notorious for resisting antibiotics, posing a serious threat to public health [1]. Therefore, the Centers for Disease Control and Prevention (CDC) defined carbapenem-resistant Enterobacteriaceae as an “urgent threat”. In the United States, resistant pathogens already annually induce 2.8 million infections [2]. Moreover, ventilator-associated pneumonia, complicated intra-abdominal infection, and bloodstream infection induced by MDR bacteria are associated with a considerable morbidity and mortality rate [3,4], placing a complex challenge for clinicians as new antibiotic development lags behind increased resistance rates.

Since the 2000s, polymyxins have been re-introduced for clinical use and are considered the last resort for MDR Gram-negative bacterial infections due to their effectiveness [5,6]. The outer membrane of Gram-negative bacteria is the target of polymyxins; the α, γ-diaminobutyric acid residue of positively charged polymyxins forms an electrostatic interaction with negatively charged phosphate groups of lipid A on the outer membrane of a Gram-negative bacterium [7]. As a result, the lipopolysaccharide is destabilized, increasing the permeability of the bacterial membrane, and resulting in cytoplasmic leakage and, ultimately, cell death [8,9]. Polymyxins can also bind to and neutralize lipopolysaccharide released during bacterial lysis, thereby preventing endotoxin reactions [10]. However, polymyxins have been gradually phased out in favor of new antibacterial medications owing to their limited antibacterial spectrum, nephrotoxicity, and neurotoxicity [11,12]. As polymyxins are antibiotics that were initially used in the 1950s, they were not subjected to modern drug development procedures [8]. Data on their pharmacokinetic and pharmacodynamic characteristics are limited, including information on their clinical applications [13]. Two types of polymyxins are currently used in humans: polymyxin B (PMB) and polymyxin E (colistin), although most of the clinical experience was derived from colistin or its prodrug, colistimethate (CMS). Extensive case–control and randomized controlled trial (RCT) studies use CMS [14,15,16]. However, limited information exists regarding PMB in clinical practice.

In 2019, several international institutions and academic organizations published the *International Consensus Guidelines for the Optimal Use of Polymyxins* [17]. Therapeutic drug monitoring (TDM) and adaptive feedback control (AFC) are recommended wherever possible for use of both colistin and PMB. An area under the plasma concentration–time curve across 24 h at a steady state (AUC_ss,24h_) of 50–100 mg·h/L is acceptable for the pharmacokinetic/pharmacodynamic (PK/PD) therapeutic targets to maximize efficacy for PMB. This corresponds to an average steady-state plasma concentration (C_ss,avg_) of 2–4 mg/L [17].

The limited sampling strategy (LSS) is useful for predicting AUC values and has recently been applied to several antibiotics in clinical practice [18,19,20], allowing accurate AUC prediction using less than three plasma drug concentrations per patient to establish a model. However, establishing an LSS is a time-consuming process requiring a relatively dense sampling of patients. To our knowledge, only two studies that included different characteristic populations from two clinical centers have documented information regarding PMB LSS [21,22]. Chen established PMB LSSs only in patients with severe pneumonia [21], whereas Wang established PMB LSSs in patients with MDR Gram-negative bacterial infections, excluding patients with renal replacement therapy [22]. Additionally, given the variability among different centers, the predictability of these models is frequently questioned when it is extrapolated to other clinical centers. Similar to the LSS of other drugs or other published population pharmacokinetic (popPK) models, poor predictabilities and extrapolability have been demonstrated by external evaluation [23,24,25]. Furthermore, PMB LSSs were established based on strict clinical trial designs, including precise infusion and sampling times, which may not reflect actual clinical practice. Therefore, it is crucial to comprehensively assess the cross-center of these published PMB LSS models using external datasets.

To provide reliable PMB LSSs for laboratories that are going to perform PMB TDM, this study evaluated the external predictability of published PMB LSSs using data collected from our center. Further, factors such as infusion and sampling time errors, which may influence PMB LSS predictability, were also analyzed.

## 2. Materials and Methods

### 2.1. Patients

A single-center clinical trial was conducted between October 2021 and July 2022 at Fujian Medical University Union Hospital. The study was performed according to the Declaration of Helsinki and its amendments and approved by the ethics committee of the Fujian Medical University Union Hospital (No.2021KJT052). Written informed consent was obtained from the patient’s relatives. Data on demographic characteristics and routine laboratory examinations were collected from electronic medical records. 

As shown in the flowchart (Appendix A), a total of 57 patients were enrolled for eligibility assessment. Finally, 27 patients were excluded, and 30 eligible patients were included in this study. The inclusion criteria were as follows: (1) aged ≥ 18 years; (2) patients who received intravenous PMB (sulfate; PMB injection, Shanghai First Biochemical Pharmaceutical Co., Ltd., Shanghai, China) after clinical diagnosis; and (3) patients who received intravenous PMB every 12 h for ≥2 days. Subjects were excluded if (1) blood sampling was unavailable after the prescribed PMB was administered; (2) they stopped PMB or died before PMB treatment; and (3) they started receiving or stopping renal replacement therapy during sampling.

### 2.2. PMB Administration and Sample Collection

According to the manufacturer’s instructions, the PMB loading dose is 100–150 mg (1 mg = 1 million units) in clinical practice. The maintenance dose was 40–75 mg every 12 h, and infusion time was approximately 1 h (±5 min error) in this study. The PMB treatment regimen, including the dose and duration of therapy, depended on the medical team. When the PMB concentration in the plasma reached a steady state (after at least four doses), samples were collected into EDTA tubes for each patient at 0 (before administration, C_0_), 1 (immediately after the end of infusion, C_1_), 2 (C_2_), 4 (C_4_), 8 (C_8_), and 12 (C_12_) h from these ICU patients via a peripherally inserted central catheter or median cubital vein. All samples were centrifuged at 9600× *g* for 5 min. The supernatant was collected and stored at −80 °C until analysis. 

### 2.3. Quantification of PMB Concentration in Plasma

As polymyxin B1 and B2 structures, molecular weights, pharmacological activities, and pharmacokinetic properties are identical, their plasma concentrations were summed to obtain total PMB concentrations [21,22]. Plasma PMB concentrations were determined using liquid chromatography–tandem mass spectrometry (LC-MS/MS, Shimadzu Jasper^TM^ HPLC system coupled to an AB SCIEX Triple Quad^TM^ 4500MD-ESI mass spectrometer, Singapore) according to a previously described method with minor modifications [26]. The calibration curves showed acceptable linearity, ranging from 0.156 to 10 μg/mL for polymyxin B1 and 0.0156 to 1.0 μg/mL for polymyxin B2. The accuracy of intraday and interday studies ranged from 80.6 to 114.9%. The coefficient of variation ranged from 2.6 to 14.8%. The plasma stability and the freeze–thaw cycle met the analytical requirements, and the methodology proved stable and dependable.

### 2.4. Development of LSSs in Our Center

Thirty complete pharmacokinetic curves were obtained from the 30 patients enrolled in the study. The pharmacokinetic curves were randomly divided into two groups: 20 for the PMB LSS modeling datasets and 10 for the validation datasets. The observed AUC_ss,24h_ (AUC*_obs_*) was calculated from all measured concentration–time points using the linear trapezoidal rule [27,28]. SPSS software (version 25.0, Inc., Chicago, IL, USA) was used to analyze the modeling group data. As previously described [21,22,29], the multiple regression method (MLR) was used to develop PMB LSSs. AUC was the dependent variable, whereas the concentration at each time point was the independent variable. Considering clinical LSS feasibility, a maximum of four concentrations was used. The relationship between AUC*_obs_* and the concentrations at each time point was analyzed using the stepwise forward method. The simplified formula for estimating PMB’s AUC_ss,24h_ (mg·h/L) was derived from the linear regression equation (Equation (1)).
(1)$${\mathrm{AUC}}_{\mathrm{ss},24h}\left(\mathrm{mg}\cdot h/L\right)=\mathrm{intercept}+\beta_{1}\times C_{t1}+\beta_{2}\times C_{t2}+\beta_{3}\times C_{t3}+\cdots +\beta_{i}\times C_{ti}$$
where β*_i_* is the fitted constant associated with each timed concentration and C*_ti_* (mg/L) is the PMB concentration at sampling time *ti*. The coefficient of determination (r^2^) was used to evaluate the equation’s regression level, and only the top five LSSs with the best r^2^ within the same concentration–time points strategy were considered for validation.

### 2.5. Validation of the Predictive Performance of LSSs Developed in Our Center

The remaining 10 PMB pharmacokinetic profiles were used to validate the developed LSSs. The predicted PMB AUC_ss,24h_ (AUC*_pred_*) was compared with the AUC*_obs_*. The Pearson correlation coefficient (R) was used to evaluate the correlation between AUC*_pred_* and AUC*_obs_*. AUC*_pred_* was estimated and compared with the corresponding AUC*_obs_* by estimating the relative prediction error (PE%, Equation (2)). There are two main criteria for evaluating predictions: bias and precision [30]. Bias is the systematic error and tendency of consistently over- or under-estimating the parameter. Precision is a random error that reflects the magnitude of the variation in the prediction. The mean prediction error (MPE, Equation (3)) was used to assess bias. The absolute precision was measured using the root mean squared prediction error (RMSE, Equation (4)). The bias and precision within ±15% were considered satisfactory and clinically acceptable [29,30,31,32]. We also applied F_15_, which indicated the percentage of PE falls within ±15%, as a combined bias and precision predictor [23,31]. If the MPE and RMSE meet the requirements, the LSS with a larger F_15_ value is preferred. For the best match LSSs, Bland–Altman plots were used to evaluate the agreement between the AUC*_pred_* and AUC*_obs_* for the highest predictive performance in each group, and the fixed range was defined as the mean ± standard deviation (SD) [33]. Data analyses and processing of graphics were performed using Excel 2016 (Microsoft Corporation, Redmond, WA, USA) and R (version 4.1.1, http://www.r-project.org, accessed on 20 October 2021).
(2)$${\mathrm{PE}}_{i}=\frac{{\mathrm{AUC}}_{pred}-{\mathrm{AUC}}_{obs}}{{\mathrm{AUC}}_{obs}}\times 100\%$$
(3)$$\mathrm{MPE}=\frac{1}{N}\sum^{\;}\left({\mathrm{PE}}_{i}\right)$$
(4)$$\mathrm{RMSE}=\sqrt{\frac{1}{N}\sum^{\;}{\left({\mathrm{PE}}_{i}\right)}^{2}}$$
where PE_i_ = prediction error, N = number of data points.

### 2.6. Validation of the Predictive Performance of LSSs in Other Study Centers

A comprehensive literature search for published PMB LSSs up to July 2022 was performed using the PubMed and Web of Science databases. The search terms were “limited sampling strategy”, “therapeutic drug monitoring”, and “polymyxin B”. The reference lists of the identified articles were manually inspected for further relevant studies. Published studies were included in the evaluation if (1) LSSs were established using MLR analysis, (2) the identified studies were performed based on LC-MS/MS for PMB quantification, and (3) the study language was limited to English. The exclusion criteria were non-intravenous administration of PMB and studies with overlapping data or cohorts; only the most recent period or the largest sample size was included.

All PMB concentration profiles from the 30 patients in our center were used as the external validation data. The PMB AUC*_pred_* was calculated using the corresponding sampling time concentration measurements within the identified LSS equations. If the concentration–time points specified by the LSSs were inconsistent with those in our study, the corresponding sampling time concentration was estimated using linear interpolation from the two adjacent measurements [34]. The predictive performance was evaluated as described using MPE, RMSE, R, and F_15_.

### 2.7. Prediction of the Performance of LSSs at Infusion Time and Sampling Time Error in Real-World Clinical Practice

#### 2.7.1. PopPK Model Analysis

As the LSSs were developed based on a relative 1 h infusion time (1 h ± 5 min error) and precise sampling, infusion time and sampling time errors are inevitable in routine clinical practice. Hence, it is necessary to explore the influence of different infusion times and sampling time errors in routine clinical practice on the LSS predictive ability. 

A popPK model was used to simulate individual serum concentration profiles for each infusion regimen [35,36]. The popPK was developed and fitted to the PMB concentration–time data using a nonlinear mixed-effect modeling approach with Phoenix NLME, version 7.5 (Pharsight, Mountain View, CA, USA). Based on previous studies [22,37,38,39,40], the first-order conditional estimation-extended least square method (FOCE-ELS) was used to develop the popPK model. The Akaike information criterion (AIC) and Bayesian information criterion (BIC) were used to set the base model. The interindividual variability of PK parameters was described by an exponential error model. Residual variability was selected with an additive error model, proportional error model, and combined error model. The covariates considered for the modeling included age, sex, total body weight (TBW), height (HT), total bilirubin (TBIL) and protein (TP), alanine aminotransferase (ALT), aspartate aminotransferase (AST), glutamyl transpeptidase (GGT), serum creatinine (Scr), and creatinine clearance (CrCl). The median of the continuous covariate was used to normalize the covariate, and the categorical covariates entered the model as power functions, with a separate dichotomous (0, 1) covariate serving as an on-off switch for each effect. A stepwise method was used to screen the covariate. A reduction in objective function values (OFVs) of >3.84 (*p* < 0.05) was considered to be statistically significant for the inclusion of one additional parameter in the forward inclusion steps. An increase in OFVs of >6.63 (*p* < 0.01) was considered to be statistically significant in backward elimination steps.

Goodness-of-fit plots were used to evaluate the final popPK model, which included observed concentrations (DV) versus population predicted concentrations (PRED) or individual predicted concentrations (IPRED), and conditional weighted residuals (CWERS) versus time (IVAR) or PRED. Moreover, a prediction-corrected visual predictive check (pc-VPC) with 1000 replicates was used to assess the model performance. The precision and robustness of parameters were assessed using the bootstrap method with 1000 datasets, which were generated using the resampling method.

#### 2.7.2. Infusion Time Error

Each patient’s PK parameters were estimated using the popPK model to simulate individual serum concentration profiles for infusion times of 0.5, 1.5, 2, and 2.5 h. The prediction qualities were assessed by calculating the MPE, RMSE, R, and F_15_ for each infusion time regimen’s predictive performance. 

#### 2.7.3. Sampling Time Error

The predictive performance of sampling time error within half an hour was evaluated in this study. Specifically, we evaluated the influence of different sampling time errors (±5, ±10, ±15, ±20, ±25, and ±30 min) on AUC*_pred_* under a 1 h infusion. The LSS (C_0_, C_1_) developed by our center was used to evaluate sampling time error on the predictive performance. Concentrations at each sampling time point were calculated based on the individual’s respective PK obtained using the above popPK modeling.

## 3. Results

### 3.1. Patients and Data Collection

A total of 180 blood samples were obtained from the 30 patients. The demographic characteristics and laboratory data are detailed in Table 1. Most patients were males (70%), and the mean age and TBW were 58.86 and 58.73, respectively. Each patient’s PMB concentration–time profiles are shown in Figure 1B. The PMB pharmacokinetic process in patients can be described using a two-compartment model. To develop and validate the PMB LSSs in our center, 30 patients were divided into the modeling (20 patients) and validation (10 patients) groups. The demographic characteristics and laboratory data did not exhibit any significant differences between the two groups (*p* > 0.05). As shown in Figure 1A, the mean PMB concentration–time curves exhibited the same trend in the model and validation groups, significantly increasing between 0 and 1 h during infusion time, reaching a peak at 1 h, falling rapidly between 1.5 and 4 h at the distribution phase, and then decreasing slowly after 4 h at the elimination phase. For the AUC*_obs_* values for the 30 patients, the median AUC*_obs_* value was 46.10 mg·h/L. However, we determined that the AUC*_obs_* values of 17 of the 30 (56.67%) patients were below 50 mg·h/L, those of 2 (6.67%) were above 100 mg/L·h, and those of 11 (36.7%) were within the acceptable target (50~100 mg·h/L). In addition, the Kolmogorov–Smirnov test was used to compare the difference between the mean PMB concentration–time curve profiles of the two groups (*p* = 0.931), which indicated no significant difference between the groups.

### 3.2. Development and Validation of LSSs in Our Center

All possible regression equations consisted of one to four concentration–time points, which were shown in Appendix A. For the LSSs with single concentration–time points, C_2_, C_4_, and C_8_ presented good correlations with AUC*_obs_*, with r^2^ values of 0.943, 0.935, and 0.922, respectively. Of the 15 LSSs that included two concentration–time points, the correlations were generally good (r^2^ > 0.900), except for LSS (C_0_, C_12_) with r^2^ = 0.896. The LSS including C_2_ and C_8_ exhibited the best correlation with an r^2^ value of 0.993. All LSSs that included three and four concentration–time points generally displayed greater correlation, with r^2^ values >0.950, except for LSS (C_0_, C_8_, C_12_) with r^2^ = 0.931; LSS (C_1_, C_4_, C_8_, C_12_) had the highest r^2^ value (up to 0.999).

The top five LSSs with the best r^2^ were determined using the validation group within the same concentration–time point strategy; their corresponding predictive performances are shown in Appendix A. Concerning the PE% for each LSS displayed in Figure 2, in the single, two, three, and four concentration–time point schemes, the C_4_, (C_0_, C_1_), (C_1_, C_2_, C_8_), and (C_1_, C_2_, C_8_, C_12_) LSSs exhibited the best predictive performance with a bias and precision within ±15%; the R and F_15_ values were >0.99 and ≥90%, respectively. For these LSS equations, the graphs describing the correlations between the AUC*_obs_* and AUC*_pred_* are shown in Appendix A. The Bland–Altman test demonstrated that none or only one plotted difference exceeded the fixed range of the mean ± SD in each model, indicating agreement between the AUC*_obs_* and AUC*_pred_* (Appendix A). The best PMB LSSs from our center are summarized in Table 2.

### 3.3. Validation of the Predictive Performance of LSSs in the Published Literature

After searching the literature, 26 PMB LSSs were identified based on relatively intensive sampling from Chen et al. [21] and Wang et al. [22] and eventually retained. As summarized in Appendix A, both studies were from China and were conducted at a single center. The infusion time and dose frequencies of both studies were 1 h and 12 h, respectively. The daily doses were 100–200 mg and 100 mg, respectively, in both studies. In addition, the included patients were diagnosed with MDR Gram-negative bacterial infections and severe pneumonia in both studies.

The predictive performance for estimating PMB AUC*_pred_* using the 26 LSSs available in the literature is provided in Appendix A. The PE% values for these LSSs are shown in Figure 3. For the single concentration–time point, of the seven equations, Equations (3)–(6) satisfied R > 0.95; only Equations (3) and (5), consisting of C_4_ and C_6_, respectively, meet the ±15% for MPE and RMSE requirements. This provides a comprehensive evaluation of bias and precision. Equations (3) and (5) had an F_15_ value of more than 70%. Nine of the eleven LSS equations met the MPE and RMSE criteria for two concentration–time points, except for Equations (8) and (10). The R of all equations was >0.95, including an F_15_ > 70%. For three and four concentration–time points, all LSS equations met the MPE and RMSE criteria within ±15%, R > 0.95, and F_15_ > 90%, except Equation (22), in which F_15_ was 83.3%. Taking all the indicators together, Equations (3), (5), (9), and (11)–(26) in Appendix A showed satisfactory predictive performances with MPE and RMSE criteria within 15%, R > 0.900, and F_15_ > 70%. The best predictive performances were observed in Equations (5) (LSS C_6_), (16) (LSS C_4_, C_6_), (20) (LSS C_1.5_, C_4_, C_8_), and (24) (LSS C_1_, C_1.5_, C_4_, C_8_), ranging from one to four concentration–time points. The correlations between AUC*_obs_* and AUC*_pred_* are shown in Appendix A for these equations, and the Bland–Altman plot was generated as shown in Appendix A. For each of the four equations, only one or two plotted differences exceeded the fixed range of mean ± 1.96 SD in the Bland–Altman test, confirming the good agreement between AUC*_obs_* and AUC*_pred_*. The best PMB LSS from the literature are summarized in Table 3.

### 3.4. Predictive Performance of LSSs at Infusion Time and Sampling Time Error in Real-World Clinical Practice

#### 3.4.1. PopPK Model

The two-compartment base model performed better than the one-compartment model (AIC and BIC of 266 and 295 versus 291 and 320, respectively); thus, a two-compartment model was used as the base model. A proportional error was applied to evaluate the residual variability. In the covariate analysis, age, sex, TBW, HT, TBIL, TP, ALT, AST, GGT, Scr, and CrCl did not exhibit a systematic relationship with PK. No correlation between random effects was identified during modeling. The final popPK parameter estimates along with bootstrap estimates are shown in Appendix A. The goodness-of-fit plots for the final model are shown in Appendix A. The observed concentrations were consistent with PRED and IPRED, and the plots of CWRES vs. time and PRED were normally distributed. The pc-VPC is presented in Appendix A, which indicates that the prediction of simulated data matched the observed plots.

#### 3.4.2. Infusion Time Error

The predictive performance of the validated PMB LSSs from published literature (met the MPE criteria within ±20%, RMSE < 20%, R > 0.95, and F_15_ > 70%) was evaluated in the 30 patients from our center at different infusion times. Individual plasma concentration profiles were simulated according to the respective individual PK obtained from the popPK modeling (Appendix A). As shown in Figure 4 and Appendix A, the predictive performances of the 0.5, 1.5, 2, and 2.5 h infusion time regimens were compared with that of the 1 h infusion time regimen.

Of all the LSS equations, when infusion time was 0.5, 1.5, 2, and 2.5 h, no LSS met the previous criteria of RMSE within ±15%. If the criteria were extended to 20% for RMSE, only the LSS (C_0_, C_1_), (C_0_, C_4_), and (C_4_, C_12_) by Chen et al. [21] and (C_1_, C_4_, C_8_) and (C_1_, C_1.5_, C_4_, C_8_) by Wang et al. [22] showed the most stable predictive performance, whereas the other LSSs showed wide fluctuations. All five LSSs satisfied the requirements of ±15% for MPE, 20% for RMSE, F_15_ > 70%, and R > 0.900, except for F_15_ of LSS (C_0_, C_1_) by Chen et al. [21], which only reached 63.33% at the 2.5 h infusion time, and the RMSE of LSS (C_0_, C_4_) by Chen et al. [21] was 20.20% at the 0.5 h infusion time.

#### 3.4.3. Sampling Time Error

Considering the clinical practice feasibility, the evaluation was performed using the LSS (C_0_, C_1_), the most commonly used strategy exhibiting the smallest prediction error at our center. As shown in Figure 5, when the sampling time errors were 0, ±5, ±10, ±15, ±20, ±25, and ±30 min, the MPE, RMSE, and F_15_ ranged from 1.46% to −15.58%, 8.57% to 18.09%, and 50.00% to 93.33%, respectively, and the R values were all >0.980 (Appendix A).

Although the predictive performance of each model varied at different infusion times and sampling errors, feasible strategies were still identified if the acceptable RMSE was extended to ±20%. The LSSs (C_4_, C_12_) described by Chen et al. [21] and those by Wang et al. [22], (C_1_, C_4_, C_8_) and (C_1_, C_1.5_, C_4_, C_8_), presented overall acceptable predictability in terms of infusion time error (0.5 to 2.5 h). However, the LSSs (C_1_, C_4_, C_8_) and (C_1_, C_1.5_, C_4_, C_8_) proposed by Wang et al. [22] were not practical because of frequent sampling. Although the F_15_ of the LSS (C_0_, C_1_) described by Chen et al. [21] was poor (only 63.33% for the 2.5 h infusion time), it was a worthwhile strategy when the infusion time was <2 h. The LSS (C_2_, C_8_) by Wang et al. [22] and that (C_4_, C_8_) by Chen et al. [21] presented the best acceptable predictability with an F_15_ of 83.3% for the 0.5 and 1.5 h infusion time. Meanwhile, the LSS (C_4_, C_12_) presented by Chen et al. [21] exhibited the best acceptable predictability with an F_15_ of 83.3% for the 2 and 2.5 h infusion time. Table 4 summarizes the recommended PMB LSSs for a more accurate AUC prediction at different infusion times in this study.

## 4. Discussion

Like most antibiotics, the efficacy of PMB is determined by drug exposure to the infection site. Suboptimal exposure may not only lead to treatment failure but also increase the emergence of resistance. Exposure in a specific patient can be evaluated using TDM, which is a significant tool for PK/PD target-guided personalized medication and thereby improves clinical outcome, whereas currently, the relationship between clinical outcomes and TDM data is scarce for PMB. Although the AUC_ss,24h_ of 50~100 mg·h/L of PMB that was recommended by international guidelines mainly comes from in vitro or animal studies, a recent study reported that the achievement of this target of PMB was independently associated with favorable clinical outcome in patients with severe pneumonia [41]. Reaching the therapeutic target of AUC_ss,24h_, a favorable microbiological response, and complications with septic shock were independently associated with favorable clinical outcomes of PMB treatment [41]. Nephrotoxicity is the most clinically relevant and dose-limiting adverse reaction of polymyxins. Wang et al. reported that the AUC_ss,24h_ of PMB in patients with renal insufficiency was slightly higher than that in patients with normal renal function, and the AUC_ss,24h_ of PMB in patients without acute kidney injury (AKI) was significantly lower than that in patients with AKI [42]. Therefore, exposure to PMB also plays a prominent role in toxicity, such as nephrotoxicity. These findings suggested that TDM of PMB is a valuable intervention that should be introduced more widely in clinical practice.

Over the past few years, only two studies have characterized TDM strategies in predicting PMB AUC_ss,24h_ in patients with MDR Gram-negative bacterial infections. However, the predictability of the established PMB LSS was unclear when extrapolated to other clinical sites because of the center-specific nature of the two studies. Here, we developed PMB LSSs based on data from our research center, assessed the practicability of PMB LSSs established in the literature, and explored the infusion and sampling time error influence on the prediction performance to obtain the optimal time points for routine PMB TDM. To the best of our knowledge, this is the first study to systematically evaluate the predictive performance of published PMB LSSs using external validation. Moreover, only 36.67% of patients who achieved the PMB PK/PD target of 50~100 mg·h/L in our study demonstrated the clinical significance of PMB TDM. Our study will provide reliable and ready-made PMB LSSs for other laboratories performing PMB TDM. 

For the LSS method, the Bayesian technique or MLR analysis can be used. The MLR technique uses an equation developed from stepwise regression analysis based on concentrations collected at predefined times after dosing and is easier to apply than the Bayesian analysis [32]. Therefore, MLR has been widely adopted for LSS studies, including the two PMB LSS studies [21,22]. Comparing the predictive performance of the PMB LSS in the published literature with that at our center, the accuracy and precision were enhanced with increased concentration–time points. When the concentration–time points were ≥3, all validated models met the MPE and RMSE criteria within ±15%, R > 0.900, and F_15_ > 70%. The improvement of the predictive ability was limited when compared with the best LSSs C_6_ and (C_4_, C_6_) by Chen et al. [21], which consisted of only single and two concentration–time points, respectively.

Specific time point inclusion may be important, as it reflects the PMB AUC. In the LSSs established in our center, 13 of the 17 (76.5%) best-matched equations included C_8_, 10 included C_2_, and 7 included C_4_. Of all LSSs from published literature tested in this study, 10 of the 19 (52.6%) best-matched equations included C_4_, 9 included C_8_, 7 included C_0_, and 6 included C_6_. C_4_, C_6_, and C_8_ differed from the other time points obtained after the fast distribution phases (Figure 1). Furthermore, LSSs C_4_ or C_6_, (C_4_, C_6_), (C_1.5_, C_4_, C_8_), and (C_1_, C_1.5_, C_4_, C_8_) showed the best ability to predict PMB AUC, consistent with the published literature [21,22]. Therefore, the concentration–time points at 4, 6, and 8 h may greatly influence the accuracy of the prediction of PMB blood exposure. 

In addition to the relatively accurate 1 h infusion time in the studies (1 h ± 5 min error), the PMB infusion time was varied or prolonged due to inaccurate infusion rate or increased dosage administered to patients in clinical practice. We also aimed to determine the predictive ability of LSSs at different infusion time schemes used in the clinic. The predictability was highly variable and depended on two factors: (1) the number of sampling points used to estimate the PMB AUC and (2) infusion time. When compared with the 1 h infusion time, the bias and precision of most LSSs indicated a considerable fluctuation under different infusion times, indicating that accurate PMB AUC prediction using LSSs depends on strict infusion time control. 

As the predictive ability of LSSs using the MLR method is dependent on the precision of blood sample collection times, sampling time error may result in inaccurate concentrations, resulting in the risk of AUC*_pred_* calculation error. Currently, most PMB AUC*_pred_* is calculated based on the trough and peak concentrations, LSS (C_0_, C_1_), which will also be recommended as part of the consensus guidelines for PMB TDM by the Division of Therapeutic Drug Monitoring, Chinese Pharmacological Society (not published). The draft recommends that the blood specimen (C_trough_ or C_0_) should be sampled 30 min before PMB administration and another specimen (C_peak_ or C_1_) within 30 min after infusion at a steady state (not published). Given that the maximum 30 min sampling time error and PMB concentration decreased dramatically during the fast distribution phase after the PMB concentration reached a peak, the C_1_ error may significantly impact the AUC*_pred_*. In our study, the sampling time error result demonstrated that the bias and precision increased and decreased from 0 to ±30 min, respectively, but still met the MPE and RMSE criteria of ±15%, except for sampling error at −25 and −30 min. As PMB peaked 1 h after administration and gradually decreased, the predicted value was underestimated. A further novel finding was that when the sampling error time ranged from −10 to 15 min, F_15_ was maintained at 90% and began to decline after −10 or 15 min. Therefore, it is reasonable to recommend that the sampling time error should be controlled within −10 to 15 min using LSS (C_0_, C_1_).

Many authors have emphasized the high variability of performance of published LSSs when evaluated in populations different from those in which they were derived [34,36]. However, several LSSs have been proposed for use in multiple patient groups rather than the same group; Ting et al. [43] determined that the LSS developed using lung transplant recipients was also applicable to a heart transplant population. Sobiak et al. [32] observed that the application of an LSS developed using mycophenolate mofetil-treated renal transplant recipients to children with nephrotic syndrome yielded satisfactory prediction results. Similarly, the LSS developed by Chen et al. [21] based on patients with severe pneumonia was validated by the data from our center, even though the population at our center included patients with severe pneumonia and other patients with MDR Gram-negative bacterial infections, including lung transplant patients. The overall validation results indicated a good predictive performance regardless of the varying population, including patients undergoing continuous renal replacement therapy and extracorporeal membrane oxygenation that covered the entire PMB therapy time. However, for patients starting or ending continuous renal replacement therapy and extracorporeal membrane oxygenation during the sampling period, PMB LSSs may not be applicable considering the sudden change in PMB pharmacokinetic profiles, resulting in an AUC calculation error. 

The popPK model was mainly used as a fit-for-purpose model to perform concentration–time curve simulation and to provide concentrations of specific times in each patient to assess the impact of errors in sampling time points and infusion length. A popPK model with a Bayesian feedback method could also be used to estimate the PK of a subject with limited blood concentration points. Furthermore, the guidelines also recommend that AFC could be used for PMB AUC prediction [17]. However, there was no accurate verification of the prediction of exposure in the patient population. In the present study, an obvious individual variation regarding AUC was observed because of large physio-pathological variations in patients. Consistent with other studies [42,44,45], we did not discover any covariate that influences the characteristics of PMB PK by establishing the popPK model due to possible insufficient samples. The methods based on the popPK model were not used because of the limited number of patients in the present study. More PK data are needed to support population PK models with potential covariates for accurate prediction.

Commercial PMB formulations for IV administration are chemical mixtures of structurally related components. The chemical structure of PMB is shown in Figure 6. Polymyxin B1, B2, B3, and PB1-I are the primary components of PB, which differ only in the fatty acyl moiety. In the study by Wang et al. [22], the total PMB concentration was obtained from the sum of the concentrations of polymyxin B1 and B2, consistent with our study. In contrast, Chen et al. [21] determined the PMB concentration using the B1, B2, and B3 concentrations. Our results showed that some LSSs from Chen et al. also exhibited excellent predictive performance after validation using PMB concentration, which was determined using only polymyxin B1 and B2. It may indicate that the contribution of polymyxin B3 concentrations to that of PMB was negligible and insufficient to affect the total PMB plasma concentration. This result was consistent with those of studies describing polymyxin B3 concentrations [46,47]. In addition, most tested LSS equations (including those that proved superior in our cohort) were derived from concentrations measured using LC-MS/MS or ultra-performance liquid chromatography–tandem mass spectrometry. LC-MS/MS was also used in our study; however, this technique may not be used in many other centers. The applicability of these equations to populations in which PMB concentrations are measured using alternative methodologies is unclear.

In addition, the evaluation of the predictive performance of LSS equations based only on r^2^ values is insufficient. During the LSS development using the MLR method, it is clear that r^2^ is a significant indicator of the correlation between the predicted and observed values. However, r^2^ only exhibits an association and provided no information regarding the prediction bias or precision. Several authors may have concentrated on LSS with the highest r^2^, ignoring others. As shown in Appendix A, the r^2^ values of LSS C_2_ and LSS C_4_ at our center were 0.943 and 0.935, respectively. Although the r^2^ value of LSS C_2_ was slightly higher than that of LSS C_4_, the LSS C_2_ precision fell beyond the ±15% range, and the LSS C_4_ exhibited the best predictive performance at single concentration–time points. Therefore, multiple comprehensive indicators should be applied to evaluate an LSS [29,31,32].

This study had several limitations. First, all PMB LSS studies were based on the Chinese Han population; the PMB pharmacokinetic profiles in different ethnicities that may influence the predictive performance of PMB LSSs need to be further validated [48]. Second, the small sample size may result in sampling bias. Third, some sampling time points specified by the equations in the published literature were inconsistent with ours; linear interpolation from the measured concentrations was applied. Given the potential inherent errors in this process, bias and imprecise estimates may occur. Fourth, sampling time errors and infusion time errors were investigated separately with different datasets and models, and the more complicated situations that include both errors have not been explored. Finally, there is little evidence in the literature on LSS therapeutic benefits in directing the PMB administration to date. Our study did not examine the association between LSSs and clinical efficacy, and further studies are needed to determine the guiding role of PMB LSS in clinical outcomes. 

## 5. Conclusions

We evaluated and validated PMB LSSs in patients with MDR Gram-negative infections. Some of these PMB LSSs established in other centers were proven to be applicable in terms of predictive performance after validation using data from our center. The constructed LSS equations by our center may also provide a reference for other researchers if more necessary validation is needed. Furthermore, the infusion and sampling time error observed in routine clinical practice can considerably affect the predictive performance of LSSs. Accordingly, we made some suggestions and offered some countermeasures. Our work provided the optimal LSSs that could be selected to better predict PMB AUC under different clinical situations, facilitating PMB TDM in other laboratories and future PMB PK/PD-based clinical studies. Further studies are warranted to verify our findings and the guiding role of PMB LSSs in clinical outcomes. With the help of PMB LSS-based AUC_ss,24h_ prediction, future research should focus on defining optimal exposure targets in patients to determine the relationship between PMB exposure and clinical success and failure in different clinical settings.

## Figures and Tables

**Figure 1 pharmaceutics-14-02323-f001:**
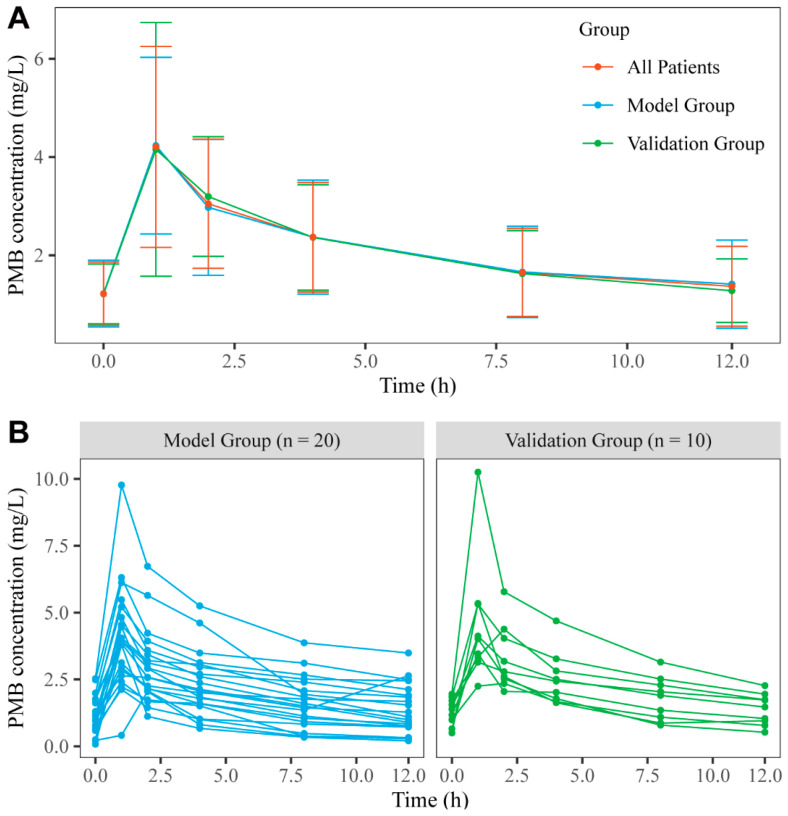
The concentration–time profiles of the model group and validation group. (**A**) The PMB concentration (arithmetic mean ± SD) versus time profiles of all patients, model group, and validation group. The red lines represent all patients, and the blue lines and green lines represent the model group and validation group, respectively. (**B**) The concentration–time profiles of the model group (n = 20) and validation group (n = 10). The blue lines and green lines represent the model group and validation group, respectively.

**Figure 2 pharmaceutics-14-02323-f002:**
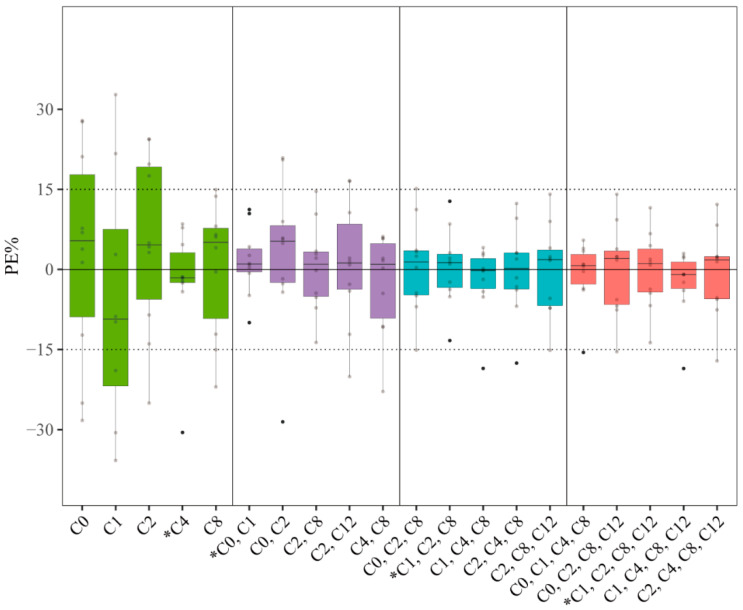
Boxplot of prediction error (PE%) for twelve models from LSSs using validation group in this study. The black solid line and dashed vertical lines represent 0% and ±15% prediction error, respectively. Green, purple, blue, and red boxes represent LSSs consisting of single, two, three, and four concentration–time points, respectively. * Model with the best predictability within the same number of concentration–time points.

**Figure 3 pharmaceutics-14-02323-f003:**
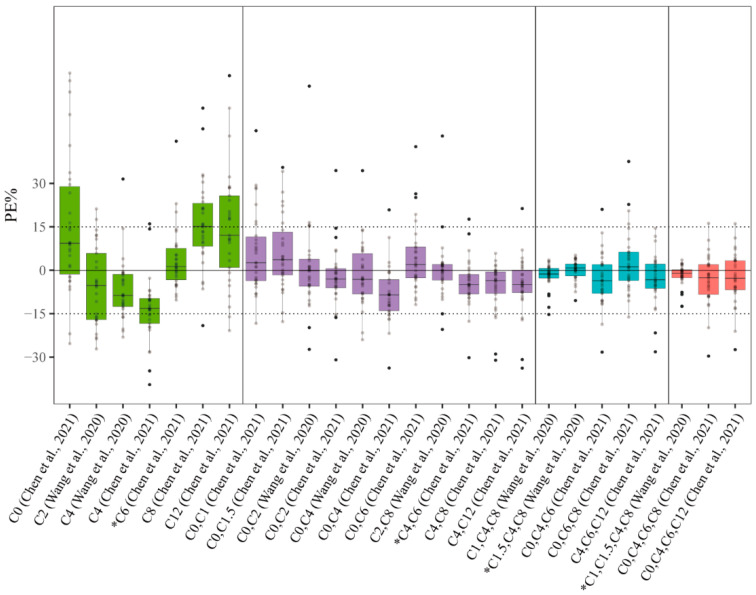
Boxplot of prediction error (PE%) of MLR-based PMB LSSs available in the literature for estimation of PMB AUC*_pred_* (Chen et al., 2021 [21] and Wang et al., 2020 [22]). The black solid line and dashed vertical lines represent 0% and ±15% prediction error, respectively. Green, purple, blue, and red boxes represent LSSs consisting of single, two, three, and four concentration–time points, respectively. * Model with the best predictability within the same number of concentration–time points.

**Figure 4 pharmaceutics-14-02323-f004:**
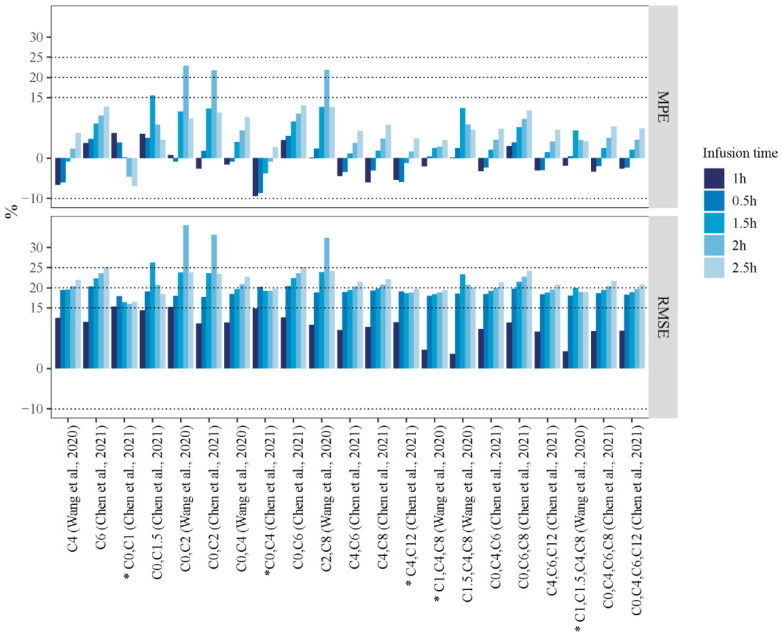
The MPE and the RMSE of the AUC*_pred_* under the selected LSSs following the different infusion times (Chen et al., 2021 [21] and Wang et al., 2020 [22]). * Model with the most stable predictive performance.

**Figure 5 pharmaceutics-14-02323-f005:**
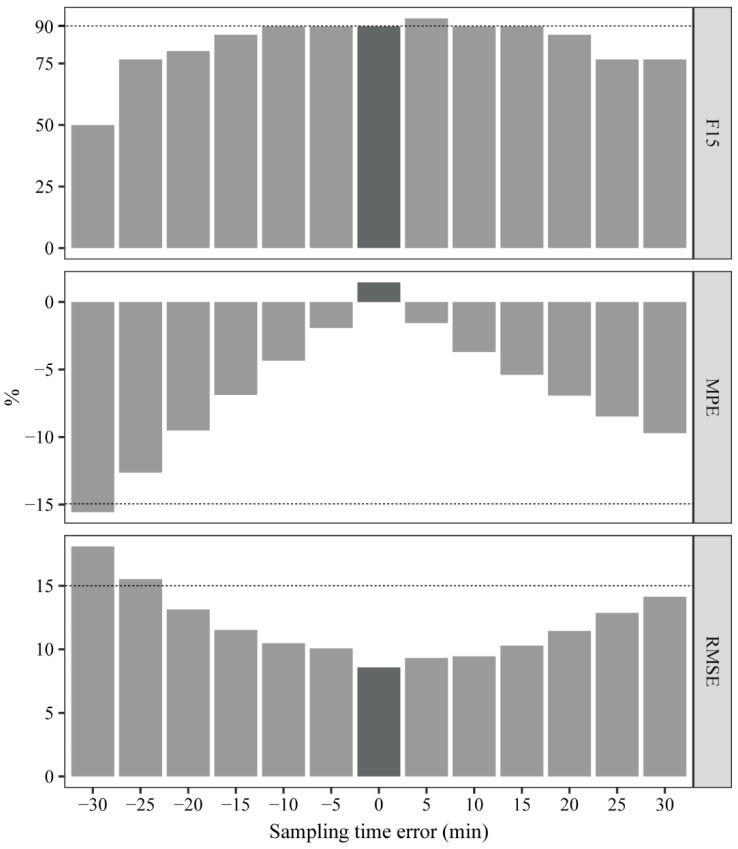
The MPE, RMSE, and F_15_ of the AUC*_pred_* using LSS (C_0_, C_1_) were developed by our center following the different sampling time errors (0, ±5, ±10, ±15, ±20, ±25, and ±30 min).

**Figure 6 pharmaceutics-14-02323-f006:**
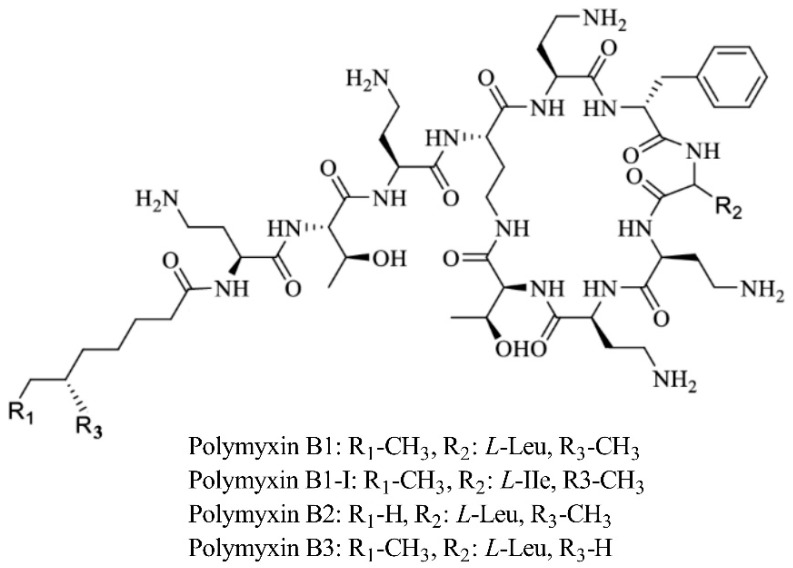
The chemical structure of PMB.

**Table 1 pharmaceutics-14-02323-t001:** Characteristics of patients included in our center.

Characteristic	All Patients (n = 30)	Model Group (n = 20)	Validation Group (n = 10)	*p* Value ^a^
	No. (%) of patients			
Sex				0.431
Male	21 (70)	15 (75)	6 (60)	
Female	9 (30)	5 (25)	4 (40)	
PMB doses (mg/12 h)				
40	1 (3.3)	1 (5)	0 (0)	0.294
50	18 (60)	10 (50)	8 (80)	
75	11 (36.7)	9 (45)	2 (20)	
Infusion duration (h)	1	1	1	
Frequency (h)	12	12	12	
	Mean ± SD or median (interquartile range)			
Age (years)	58.86 ± 17.01	59.21 ± 16.81	58.20 ± 18.27	0.882
Total body weight (kg)	58.73 ± 10.93	60.83 ± 11.55	54.55 ± 8.60	0.141
Height (cm)	166.17 ± 6.79	166.80 ± 6.88	164.9 ± 6.77	0.480
Total bilirubin (µmol/L)	47.06 ± 48.30	40.13 ± 38.78	59.53 ± 62.35	0.317
Total protein (g/L)	56.94 ± 9.15	56.19 ± 9.61	58.52 ± 8.41	0.539
Alanine aminotransferase (U/L)	35.28 ± 33.47	29.89 ± 32.40	45.50 ± 34.75	0.239
Aspartate aminotransferase (U/L)	45.17 ± 29.00	39.11 ± 19.58	56.70 ± 40.30	0.122
Glutamyl transpeptidase (U/L)	74.55 ± 85.18	71.53 ± 95.35	80.30 ± 65.59	0.797
Serum creatinine (µmol/L)	123.21 ± 83.10	126.42 ± 96.03	117.10 ± 54.5	0.780
Creatinine clearance (mL/min)	66.37 ± 45.84	69.41 ± 45.32	60.60 ± 48.72	0.631
AUC*_obs_* (mg·h/L)	46.10 (34.02–65.52)	46.09 (32.36–68.75)	47.69 (36.40–65.76)	0.971

Creatinine clearance was calculated according to the Cockcroft–Gault formula. AUC*_obs_*, the observed area under the plasma concentration–time curve across 24 h at a steady state. ^a^ Fisher’s exact test was used for comparing the proportions of categories in two group variables, and the Mann–Whitney *U* test was used to compare continuous variables.

**Table 2 pharmaceutics-14-02323-t002:** Predictive performance of the best PMB LSSs developed by MLR analysis from our center.

Time Point	Equation	MPE%	RMSE%	R	F_15_%
C_4_	20.623 × C_4_ + 2.889	−2.31	10.6	0.994	90
C_0_, C_1_	23.006 × C_0_ + 6.037 × C_1_ − 1.853	1.53	6.21	0.992	100
C_1_, C_2_, C_8_	0.600 × C_1_ + 8.356 × C_2_ + 14.078 × C_8_ + 0.997	0.44	6.9	0.992	100
C_1_, C_2_, C_8_, C_12_	0.637 × C_1_ + 8.749 × C_2_ + 9.749 × C_8_ + 4.380 × C_12_ + 0.669	−0.18	6.87	0.991	100

MPE, mean prediction error; RMSE, root mean squared prediction error; R, the Pearson correlation coefficient between AUC*_pred_* and AUC*_obs_*; F_15_, the percentage of prediction error falling within the ±15%.

**Table 3 pharmaceutics-14-02323-t003:** The predictive performance of best MLR-based PMB LSSs available in the literature for estimation of PMB AUC*_pred_* in patients with MDR Gram-negative bacteria infection treated with PMB.

Time Point	Equation	Reference	MPE%	RMSE%	R	F_15_%
C_6_	8.147 + 21.961 × C_6_	[21] ^a^	3.71	11.57	0.993	90
C_4_, C_6_	2.030 + 8.532 × C_4_ + 13.465 × C_6_	[21]	−4.34	9.48	0.991	90
C_1.5_, C_4_, C_8_	0.599 + 1.964 × C_1.5_ + 3.169 × C_4_ + 6.633 × C_8_	[22] ^b^	0.13	3.63	0.997	100
C_1_, C_1.5_, C_4_, C_8_	0.260 + 0.460 × C_1_ + 1.137 × C_1.5_ + 3.644 × C_4_ + 6.480 × C_8_	[22]	−1.84	4.26	0.998	100

MPE, mean prediction error; RMSE, root mean squared prediction error; R, the Pearson correlation coefficient between AUC*_pred_* and AUC*_obs_*; F_15_, the percentage of prediction error falling within the ±15%. ^a^ Prediction of AUC_ss,24h_. ^b^ Prediction of AUC_ss,12h_.

**Table 4 pharmaceutics-14-02323-t004:** Predictive performance of PMB LSSs from the published literature which was validated in our center and LSSs recommended for routine clinical practice.

Infusion Duration	Time Point	LSSs Equation	Reference	MPE%	RMSE%	R	F_15_%
0.5 h	C_2_, C_8_	−0.274 + 4.671 × C_2_ + 7.181 × C_8_	[22] ^a^	2.31	18.83	0.947	83.33
1 h	C_6_	8.147 + 21.961 × C_6_	[21] ^b^	3.71	11.57	0.993	90.00
	C_4_, C_6_	2.030 + 8.532 × C_4_ + 13.465 × C_6_	[21]	−4.34	9.48	0.991	90.00
1.5 h	C_4_, C_8_	0.196 + 13.903 × C_4_ + 9.725 × C_8_	[21]	1.90	19.85	0.942	83.33
2 h	C_4_, C_12_	0.546 + 14.120 × C_4_ + 11.235 × C_12_	[21]	1.63	18.86	0.947	83.33
2.5 h				4.93	19.67	0.948	83.33

MPE, mean prediction error; RMSE, root mean squared prediction error; R, the Pearson correlation coefficient between AUC*_pred_* and AUC*_obs_*; F_15_, the percentage of prediction error falling within the ±15%. ^a^ Prediction of AUC_ss,24h_. ^b^ Prediction of AUC_ss,12h_.

## Data Availability

All data included in this study are available from the corresponding author upon request.

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
