# Peer review of "Evaluation and Validation of the Limited Sampling Strategy of Polymyxin B in Patients with Multidrug-Resistant Gram-Negative Infection"

_pharmaceutics, 2022, doi:10.3390/pharmaceutics14112323_

Round 1

Reviewer 1 Report

The manuscript is a well-written study that demonstrated the validated Polymyxin B (PMB) -limited sampling strategy based on prediction of AUC values in patients with Multidrug-Resistant Gram-Negative Infection.

The data provided meaningful information of the possiblity for PMB therapeutic drug monitoring (TDM) and clinical usefulness. 

However, there are several minor aspects that need to be clarified by the authors:

1. In this study, a total of 30 patients was enrolled; : 20 for the PMB LSSs modeling datasets and 10 for the validation datasets.

In my oipnion, the number of patients involved in popPK model was relativley small. Were 20 patients sufficient to show the conclusion?

2. The TDM monitoring or pharmacokinetics in patients could be valuable when it provided clinical benefit.  

Although it was limited, it was necessary to discuss the results of this study with the relation of clinical outcomes such as laboratory results, hospitalization period or death rate.

3. Polymyxins have been used for limited purpose due to their safety profiles. Please, explain safety profiles or issues including nephrotoxicity especially in in patients with high drug concentrations

Author Response

We are grateful to the reviewers for their comments and suggestions, which have helped us to improve our manuscript.  Please find the point-by-point responses in the attachment.

Reviewer 2 Report

The manuscript has clearly figured out the hypothesis and is suitable for publication in this journal. However, If it possible authors please add below points in the revised manuscript

  1. Polymyxin B (PMB) Structure and Mechanisms of Action Against Multidrug-Resistant Gram-Negative Bacteria
  2. Add some more points about the multidrug-resistant gram-negative bacteria in the introduction part.
  3. Rewrite conclusion with future prospective 

Author Response

We are grateful to the reviewers for their comments and suggestions, which have helped us to improve our manuscript.  Please find the point-by-point responses to the reviewer’s comments in the attachment.

Reviewer 3 Report

The manuscript “Evaluation and Validation of the Limited Sampling Strategy of 2 Polymyxin B in Patients with Multidrug-Resistant Gram-Negative Infection “ by Li and coworkers was reviewed for publication in the Pharmaceutics. During the study of the presented work, I found major problems that should be carefully addressed to achieve a meaningful contribution to the scientific literature.

The following text contains questions and comments regarding the different sections of the presented work.

I Introduction

1.       The authors state (as in the Abstract) that for maximum efficiency 50—100 mg*h/L are “required” and cite the consensus guidelines published by Tsuji and coworkers. The guideline clearly states that the recommended target is 50 mg*h/L (see R3 and especially the summary with “Therefore, the panel recommends the same target exposures as for colistin (AUCss of~50 mg*hour/L)”).

II Methods

Clinical Trial and Data Acquisition:

1.       Please provide a Flowchart for the clinical trial with information on cardinality for each stage and recruitment/enrollment/inclusion/exclusion criteria.

2.       It is not clear how many patients were excluded (30 were enrolled and the PK in all 30 patients was investigated).

3.       What does “±5 min error” in Section 2.2 exactly mean? In 2.71 the authors mention “an accurate 1h infusion time”.

Statistics and LLS Linear Models:

1.       Pharmacokinetic data (e.g. AUC) is typically log-normal distributed. Hence, using arithmetic means and standard deviations is not appropriate (see summary statistics in  Table 1).

2.       If baseline characteristics are tested, it is not clear why Sex (e.g. % Female) was not tested (e.g. by Fisher exact test).

3.       It is unclear what the p-column in Table 1 means. Is it a p-value (as the uppercase “P” in the text)? If this is the case, which test was used to investigate baseline imbalances?

4.       For the LLS equation, units for AUC (output) and concentrations (predictor) should be stated somewhere in the text.

5.       Please use standard and correct wording for statistical terms (e.g. not “partial correlation coefficient” for regressor weights or “regression coefficient”).

6.       The sentence ending with “… only good regression equations were considered for the validation” is extremely vague. Please be precise.

PopPK Modeling and Simulations:

1.       Please be precise with the wording here. FOCE-ELS was not used to “compare” models.  FOCE-like algorithms are used for the ML estimation step. In 2.7.2 individual parameters are not “generated” – they are (presumably) MAP-estimated from the specifications of the final model.

2.       Median normalization for an (assumed) multiplicate covariate modeling was conducted for the (assumed) continuous covariates. What was used for the dichotomous variable “sex”? Please be precise here.

3.       It is not clear why covariate exploration was done in the first place since the model was only used as a fit-for-purpose model.

III Results:

1.       In Section 3.1 the authors state that “…the mean PMB concentration-time curve profiles and AUCobs were not significantly different between the groups”. Which tests were used and did the authors test the difference between the profiles and not only AUC? Please report p-values instead of stating the relation to your chosen alpha level.

2.       In Section 3.4.1 the authors present the “final popPK model equations”. These are the parameters and not the equations. Moreover, the authors present an IIV on CL2 but there is no IIV on CL2 in Table S3 but CL2 in Table S4 seems to be individually estimated. Since the model has IIV on all parameters (and even if it is just fit-for-purpose) the authors should provide information on shrinkage to get an idea regarding over-parametrization.

3.       In 3.4.2: Do the authors mean “validated PMB LSSs” for their dataset?

4.       It is very confusing to understand the rationale in 3.4.2? Why did the authors not also test their own LSSs for varying infusion times? The reader cannot easily compare Table 3 references with Figure 4 references to even understand/compare the baseline 1h values.

5.       Since the authors characterized the PK via a population approach it seems counter-intuitive to not sample from the distributions for their simulations and instead only use the MAP-estimated 30 individuals (especially since no covariate effects were found).

6.       It is not clear at all how sampling time errors were investigated. Was it always +X minutes for every time point or was it randomized within an interval (e.g. +- X minutes; that would make more sense)?

7.       At the end of the Results, the authors summarize the infusion times modeling (external LSSs vs. their data) and mention the sampling time modeling (their LSSs vs. their data). This is so confusing since these are very different comparisons (extrapolated external evaluations vs. some extrapolated internal validation for different investigations).

8.       R^2 (as e.g. in Table 2) is not the Pearson correlation coefficient (as it is not squared). It is likely the coefficient of deterioration (unclear if adjusted or not). In Table 3 we see the same error but since this is only a model application, it is even more confusing which parameter it is (likely the square of the Pearson correlation coefficient).

IV Discussion:

1.       The authors claim that their cohort had “considerably low AUCobs”. This should be interpreted with respect to the precise guidelines since the problem might also be the considerable fraction of quite high AUCobs values (> 50mg*h/L).

2.       What does “significant” in “During the validation process, there was no significant difference in LSS predictive performance, …” (lines 432—433) mean?

3.       To this point in the manuscript, it is not clear if the authors describe the Pearson correlation coefficient or the (adjusted) r^2 when they mention r^2 (see Table 2/3 for the confusion).

4.       The finding, that r^2 is not a metric for e.g. bias is textbook knowledge and, thus, irritating that this is presented as a “finding”. Furthermore, the authors state that “Most authors have concentrated on time points with the highest r2, ignoring others.” Please provide the evidence for studies where the authors only cared about the r^2?

5.       The PopPK model is not discussed at all.

6.       Please use the term “ethnicities” or “populations” in line 455.

7.       The limitation section should at least include the limitation that sampling time errors and infusion time errors were investigated separately with different datasets and models (still not sure why) and this is certainly not the case for clinical practice (a combination of both errors).

8.       At the end of the manuscript (Conclusion) it is not clear why the authors constructed their own LSS equations if they wanted to externally validate the LSSs of other researchers with their own data.

V Presentation

Tables and Figures:

1.       Figure 1 composition is not ideal (recommendation: Model and Validation in the bottom row and summary plot in the top row). The Figure caption is missing information about the type of averaging (arithmetic mean?) and the description of the error bars (SD?).

2.       Tables 2 and 3 should not be placed in the main manuscript. The best combination of concentration-time points per group could be presented (Table 1) with the necessary performance metrics. The equations with regressor weights should also not be presented in the main manuscript since these do not provide any meaningful information if not “used” for application.

3.       Figure 3 does not allow the comparison of the LSSs from the two different studies (Chen and Wang) since it is ordered by time of measurement.

VI Supplementary Information:

1.       Please add units to the parameter values in Table S3 and state the type of residual error model. The final error model is nowhere stated in the manuscript or supplementary information.

VII General Writing and Presentation:

1.       Typography needs substantial editing work (e.g. missing whitespace).

2.       Consistency in wording should be improved (e.g. “Eq. X” vs. “Equation X”).

3.       Writing and language must be improved (there are even a couple of missing words in sentences).

4.       The resolution of the Figures is too low.

Author Response

(The authors gave the same response as above.)

Round 2

Reviewer 3 Report

The reviewer thanks the authors for adequatly addressing all comments and questions.